# Surrogate-Model-Based Interval Analysis of Spherical Conformal Array Antenna with Power Pattern Tolerance

**DOI:** 10.3390/s22249828

**Published:** 2022-12-14

**Authors:** Guangda Ding, Peng Li, Paolo Rocca, Jiantao Chang, Wanye Xu

**Affiliations:** 1The Key Laboratory of Electronic Equipment Structure Design, Ministry of Education, Xidian University, Xi’an 710071, China; 2The ELEDIA Research Center, ELEDIA@UniTN—University of Trento, Via Sommarive 9, 38123 Trento, Italy

**Keywords:** interval arithmetic, surrogate model, spherical conformal array antenna, tolerance analysis

## Abstract

A method based on an interval arithmetic is proposed to analyze uncertain factors such as the curvature radii, excitation amplitude, and excitation phase of a spherical conformal array antenna. An interval description of element factors under different curvature radii of spherical substrates is established using the surrogate model based on the data obtained through a full-wave analysis method. The interval formula of the spherical curvature radius and array element position error is derived and the effects of the spherical radius tolerance, excitation amplitude tolerance, and excitation phase tolerance on the antenna power pattern are studied. To evaluate the effectiveness and reliability of the proposed method, a set of representative numerical results are reported and discussed and a comparison with the Monte Carlo methods and full-wave simulation is described. This method can be widely used during the antenna design and before the antenna prototyping/manufacturing to predict the effects, on the radiation performance, of possible errors/tolerances in the antenna structure to guarantee the antenna working ‘in operation’.

## 1. Introduction

Conformal array antennas (CAAs) refer to an array antenna attached to and fitted to a carrier surface, i.e., a non-planar conformal antenna array that needs to be conformally installed on a fixed-shape surface [1]. The patch array antenna is widely used in aerospace [2,3] and military [4] communication equipment owing to its high gain characteristics, flexible shape, controllable beam shape, and ease in conforming to the surfaces of dielectric substrate.

To meet the aerodynamic requirements of aircraft, CAAs usually have a special shape that incorporates the shape of an airborne radome [5]. Compared with array antennas, whose traditional plane configurations severely limit their application in aircraft with curved shapes, CAAs have the feasibility of a joint other physical geometry, space saving, increased array aperture, and reduced aerodynamic drag, greatly improving the space utilization and stealth performance of a carrier [6,7,8]. Therefore, the realization of conformal array antennas has significant potential in future applications, which has attracted the attention of scholars [9]. Ma et al. designed and manufactured a curved dielectric plate antenna that can provide a wider bandwidth and a higher gain [10]. Ramadan et al. also studied a wearable conical antenna and controlled its RF performance while maintaining the flexibility of the antenna [11]. With the wide application of conformal antennas, 3D printing technology provides a channel for the processing of antenna models. As a disadvantage, the accuracy is slightly inferior to that of the traditional chemical etching method [12]. Therefore, it is necessary to explore the influence of uncertain factors of the antenna structure on its power pattern.

Uncertain factors of array antennas, mainly structural and system factors, affect the actual performance of the antenna. Therefore, an interval analysis of the electrical performance can be carried out according to uncertain factors of the antenna. There are two main methods of interval analysis: probability- and improbability-based research methods [13,14]. Based on the research method of probability, Ruze was the first to study the influence of aperture tolerance on antenna radiation patterns, taking the uncertainty variable as a Gaussian distribution for research by analyzing a large number of statistical data [15,16]. Gilbert studied the influence of random geometric errors on antenna pattern gain [17]. Elliott analyzed the influence of random errors in the structure and excitation on the sidelobe level; this method can only analyze the influence of random errors on a certain electrical performance [18]. Samii proposes a research based on Ruse that considers non-uniform root mean square surface error and non-uniform illumination function at the same time to determine the influence of different random errors and illumination taper on parameters such as gain loss and sidelobe level [19,20]. Hsiao used statistical methods to study the relationship between random error and maximum sidelobe level, which is of great significance in radar systems [21,22]. Ling studied the probability distribution of reflector antenna sidelobe levels affected by some random surface errors [20]. With the development of the antenna model toward a higher precision, this method requires a lot of time and numerous resources. Therefore, more researchers now choose such methods based on probability.

Research methods based on improbability mainly consider interval arithmetic (IA) and have been well developed in recent years [23,24,25,26]. In general, the upper and lower bounds of the variables are easily obtained through a small number of experiments and, thus, the interval arithmetic treats the uncertain factors as interval variables. Rocca et al. exploited an interval analysis method to explore the influence of the uncertainties of array factors on the power pattern of an array antenna, including the excitation amplitude tolerance [27,28,29] and the excitation phase tolerance [30,31]; when the excitation distribution is known, the variation range of the radiation pattern is deduced by the interval expression. Later, Anselmi et al. studied the influence of the excitation amplitude and phase errors on the radiation pattern of the array antenna based on the interval center and radius [28,30], which laid a foundation for a subsequent interval analysis. Li et al. used interval analysis arithmetic to study the influence of radome uncertainties on the far-field pattern, including the radome thickness [32] and permittivity of the materials [33]. Wang et al. studied the tolerance of a 3D-printed patch antenna and proved that the influence of the element and array factors on the power pattern cannot be ignored [34]. 

To the best of the authors’ knowledge, (1) considering only the tolerance analysis of planar array antennas in the existing works, this study focuses on the effects of the structural and excitation tolerance on the power pattern of spherical CAAs; (2) considering only the tolerance of the array factor, the tolerances in both element factor and array factor could be researched by this study. Based on the IA method, according to the geometric characteristics of the sphere, the formula of the element position, excitation amplitude, phase, and pattern of the CAAs are derived. The interval of the element factors under different radii of a spherical substrate is established using a surrogate model to describe the data obtained by the full-wave analysis method. Finally, the influence of the uncertain factors of the CAAs on the antenna power pattern is studied based on the element and array factors.

The remainder of this paper is organized as follows. The theoretical formula for a spherical CAA is derived in Section 2. Some numerical examples are used to verify the validity and accuracy of the proposed method in Section 3. Section 4 describes the analyses and studies the uncertain factors of the CAA. Finally, some conclusions are drawn in Section 5.

## 2. Materials Formula

For linear array and planar array, the beam direction of each directional element is the same if the same element is used. According to the pattern product theorem [35], the array pattern can be expressed in the form of the product of element factors fa and array factors fe, as shown in (1).
(1)F(θ,φ)=fa(θ,φ)⋅fe(θ,φ)

Since the elements of conformal array are distributed on the surface of the carrier and the direction of the pattern of each directional element is different, the multiplication principle similar to the linear array cannot be simply used to solve the pattern. The pattern of conformal array antenna will be analyzed below.

As shown in Figure 1a, for N-element arbitrary antenna array in space, let the position of the nth array element at $r_{n}$ in the array be $\left(x_{n},y_{n},z_{n}\right)$ and the excitation of the array element be $w_{n}=A_{n}e^{jB_{n}}$. $A_{n}$ is the excitation amplitude and $B_{n}$ is the excitation phase. For observation point *P* in the far field (θ,φ), the direction vector is
(2)α=(cosϕsinθ,sinϕcosθ,cosθ)

With the coordinate origin *O* as the phase reference point, the path difference between the element arriving at far-field point *P* and the reference point arriving at point *P* is
(3)$$d_{n}=r_{n}\cdot \alpha =x_{n}\cos\phi \sin\theta +y_{n}\sin\phi \cos\theta +z_{n}\cos\theta$$

The radiation pattern of the nth antenna array element at point *P* can be expressed as
(4)$$F_{n}(\theta ,\varphi )=f_{n}(\theta ,\varphi )\cdot w_{n}\cdot e^{jkd_{n}}$$
where $f_{n}(\theta ,\varphi )$ is the element pattern of the nth array element. According to the superposition principle of electromagnetic field, the far-field radiation pattern of this N-element array is
(5)$$\begin{array}{cc} F(\theta ,\varphi ) & =\sum_{n=1}^{N}F_{n}(\theta ,\varphi )=\sum_{n=1}^{N}f_{n}(\theta ,\varphi )\cdot w_{n}\cdot e^{jkd_{n}}\\ & =\sum_{n=1}^{N}f_{n}(\theta ,\varphi )\cdot A_{n}\cdot e^{j(B_{n}+k(x_{n}\cos\phi \sin\theta +y_{n}\sin\phi \cos\theta +z_{n}\cos\theta ))} \end{array}$$

### 2.1. Array Factors of Spherical CAAs

The derivation of the array factor needs to start from the calculation of the coordinates of the central point of each element. Because of the perfect spatial symmetry of the spherical array structure, it can be regarded as consisting of several ring arrays, and the radius of each ring is determined by its position distribution on the sphere. Taking the m×n CAA in Figure 2 as an example (the radius of the sphere is *Rs*), where α is the angle between the two antenna elements relative to the origin of the coordinate and it is related to the length and width of the element patch antenna. $\alpha_{m}$ represents the angle between the *m*th row antenna and the *x*-axis on the plane and $\alpha_{n}$ represents the angle between the *n*th column antenna element and the *y*-axis on the *yoz* plane. $\alpha_{m}$ and $\alpha_{n}$ are angles calculated based on the number of elements and the spherical radius $\alpha_{m}=1/2(\pi -(m-1)\alpha )$ and $\alpha_{n}=1/2(\pi -(n-1)\alpha )$. Therefore, the position of the center point of the antenna element in the *m*th row and *n*th column can be calculated based on the geometric relationship of the sphere.
(6)$$\begin{array}{cc} R_{m} & =Rs\times \cos(\pi /2-\alpha_{m})\\ x_{mn} & =Rs\times \sin(\pi /2-\alpha_{m})\\ y_{mn} & =R_{m}\times \cos\alpha_{n}\\ z_{mn} & =R_{m}\times \sin\alpha_{n} \end{array}$$
where *R_m_* represents the radius on the *xoz* plane of the antenna in the *m*th row.

For spherical CAAs, all antenna elements were distributed in a three-dimensional (3D) space, as shown in Figure 1. The position of each antenna element is different in the 3D cartesian coordinate system. The far radiation field of the antenna element is the product of the vector pattern of the antenna element and the spherical wave $e^{-jk_{0}R_{mn}}/R_{mn}$. The radiation far field of the antenna element after approximate processing can be expressed as follows in spherical coordinate system [36]:(7)$$\begin{array}{cc} E_{mn}\left(\theta ,\varphi ,r\right) & =A\cdot fe_{mn}\left(\theta ,\varphi \right)\frac{e^{-jk_{0}R_{mn}}}{R_{mn}}\\ & =\frac{e^{-jkR}}{R}fe_{mn}\left(\theta ,\varphi \right)e^{jkr_{mn}\cdot \hat{r}} \end{array}$$
where $r_{mn}$ is the position vector of the element antennas in the mth row and nth column relative to the origin O of the 3D cartesian coordinate system, $\hat{r}$ is the unit vector in the P(x,y,z) direction at some point in the far field. Here, is the coordinate transformation between the two coordinate systems, $r_{mn}=x_{mn}\hat{x}+y_{mn}\hat{y}+z_{mn}\hat{z}$, $\hat{r}=u\hat{x}+v\hat{y}+\cos\theta \hat{z}$, where u=cosφsinθ and v=sinφsinθ. *A* is a constant, which is usually omitted in the analysis of array antennas. Therefore, the array factor of the CAAs obtained from the superposition theorem is as follows. It is the closed-form equation of array factor fa and spherical radius r.
(8)$$fa\left(\theta ,\varphi \right)=\sum_{m=1}^{M}\sum_{n=1}^{N}w_{mn}e^{jk_{0}\left(ux_{mn}+vy_{mn}+\cos\theta z_{mn}\right)}$$

### 2.2. Establishment of Element Factor Interval Surrogate Model Based on Machine Learning

The patch antenna is widely used as the element in array antenna. The pattern (element factor) formula of traditional planar patch antenna can be found in much of the literature [12,34]. The tolerance analysis of structure factor such as dimensions and materials based on IA has also been conducted in [34], however it cannot be used for the spherical patch antenna of CAAs due to the limitations of the theoretical formula. Therefore, this section uses the surrogate model obtained by the method of artificial intelligence to overcome the theoretical derivation of a closed-form formula for the spherical patch antenna pattern, just as the Taylor series has been used as the surrogate model to replace the rigorous theoretical deduction and reduce the overestimate of IA in [34,37,38].

The process of obtaining the surrogate model is shown in Figure 3. Firstly, the full wave analysis method (HFSS) was used to establish the element model of conformal patch antenna. Then, the interval of the spherical patch antenna pattern with a structure tolerance interval was obtained by the Monte Carlo (MC) method and the data were calculated via HFSS used for training and validation and new data predictions using a backpropagation (BP) neural network. Finally, the data were sorted and normalized and the upper and lower boundary formulas of element factors were fitted in MATLAB. Here, Rs represents the radius of the spherical substrate, the subscript inf in $Rs\in [Rs_{\inf},Rs_{\sup}]$ represents the lower boundary, and sup represents the upper boundary.

The BP neural network is a multi-layer feedforward network trained according to the error backpropagation algorithm and it is also one of the most widely used neural network models. The basic structure of a neuron is shown in Figure 4, which consists of an input layer, a hidden layer, and an output layer. Nodes (neurons) relate to some specified weights. Using the BP trained network to predict new data can not only obtain accurate data but can also save a lot of time in HFSS simulation. As shown in Figure 5, the green line represents the E-plane/H-plane pattern under different radii within a group of radius errors simulated using the BP neural network; the maximum $fe_{i\text{-}HFSS}(\theta )$ and minimum $fe_{i\text{-}HFSS}(\theta )$ values of the corresponding angle points in this group of data are proposed as the upper boundary $fe_{\sup}(\theta )$ (red line) and lower boundary $fe_{\inf}(\theta )$ (blue line) of the element factor interval. Finally, the upper and lower boundaries were fitted using MATLAB to obtain the approximate function about θ as the surrogate model of the element factor. The midpoint and radius were used to represent the element factor interval as follows [28].
(9)$$\begin{array}{cc} fe_{\sup}(\theta ) & =\max\{fe_{i\text{-}HFSS}(\theta )\}\\ fe_{\inf}(\theta ) & =\min\{fe_{i\text{-}HFSS}(\theta )\}\;i=1,2,\ldots ,n \end{array}$$
where i represents the number of simulations in HFSS, $\theta \in (-90^{{}^{\circ}},\;90^{{}^{\circ}})$ and then $fe_{mid}=1/2\times (fe_{\inf}+fe_{\sup})$ and $fe_{rad}=1/2\times (fe_{\sup}-fe_{\inf})$; here the subscripts mid and rad (the same below) represent the midpoint and radius of the interval, respectively.

### 2.3. Internal Power Pattern of Spherical CAAs

After the formula of the element factor fe(θ,φ) and array factor fa(θ,φ) of CAAs is obtained, the far-field pattern of the CAAs can be expressed as (5) and substituted into $P\left(\theta ,\varphi \right)={\left|F\left(\theta ,\varphi \right)\right|}^{2}$ to calculate the power pattern.
(10)$$F\left(\theta ,\varphi \right)=\sum_{m=1}^{N}\sum_{n=1}^{N}w_{mn}fe_{mn}\left(\theta ,\varphi \right)\times \sin(a_{n})e^{jk_{0}\left(ux_{mn}+vy_{mn}+\cos\theta z_{mn}\right)}$$

The requested steps to obtain the bounds of power pattern from the radius error intervals are shown in Figure 6. After determining the tolerance range of the radius of the spherical substrate, we can obtain the boundary of the coordinate of the center point of the antenna element according to the geometric relationship between the center point of the antenna element and the spherical radius in the cartesian coordinate system and we can substitute it and the element factor surrogate model into (5) to obtain the boundary of the far-field pattern. Then, it can be expanded into real part and imaginary part and divided into many cases to solve the boundary of power pattern.

When there is a tolerance, it is assumed that the element factor of each element antenna in the array antenna has the same upper and lower boundaries; then the interval formula for the mth row and nth column element pattern is shown in (11), where the subscript IA represents the interval value (the same as below) and $fe_{IA}\left(\theta ,\varphi \right)\times \sin(a_{n})$ indicates that the element factor interval takes the component on the *z*-axis.
(11)$$F_{mn-IA}\left(\theta ,\varphi \right)=w_{mn}fe_{IA}\left(\theta ,\varphi \right)\times \sin(a_{n})\times e^{jk_{0}\left(ux_{mn-IA}+vy_{mn-IA}+\cos\theta z_{mn-IA}\right)}$$

The radius of curvature of the mth antenna on the *xoz* plane is expressed as follows:(12)$$R_{m-IA}=(Rs\pm \delta )\times \cos(\pi /2-\alpha_{m})$$

The *x*-, *y*-, and *z*-axis coordinates of the center point of the patch antenna element in the mnth are expressed as:(13)$$\begin{array}{l} x_{mn-IA}=(Rs\pm \delta )\times \sin(\pi /2-\alpha_{m})=x_{mn-mid}\pm x_{mn-rad}\\ y_{mn-IA}=R_{m-IA}\times \cos\alpha_{n}=y_{mn-mid}\pm y_{mn-rad}\\ z_{mn-IA}=R_{m-IA}\times \sin\alpha_{n}=z_{mn-mid}\pm z_{mn-rad} \end{array}$$

The central coordinate position of the element is located in the exponent part in (11) and can be replaced by Φ*_mn_*:(14)$$\begin{array}{l} \Phi_{mn-\inf}=k_{0}\left(ux_{mn-\inf}+vy_{mn-\inf}+\cos\theta z_{mn-\inf}\right)\\ \Phi_{mn-\sup}=k_{0}\left(ux_{mn-\sup}+vy_{mn-\sup}+\cos\theta z_{mn-\sup}\right)\\ \Phi_{mn-mid}=k_{0}\left(ux_{mn-mid}+vy_{mn-mid}+\cos\theta z_{mn-mid}\right)\\ \Phi_{mn-rad}=1/2\times (\Phi_{mn-\sup}-\Phi_{mn-\inf}) \end{array}$$

The array pattern is the sum of all element patterns, as shown in (15):(15)$$F_{IA}\left(\theta ,\varphi \right)=\sum_{m=1}^{N}\sum_{n=1}^{N}F_{mn-IA}\left(\theta ,\varphi \right)$$

The interval formulas of (15) is obtained as follows via the midpoint and radius notation in interval theory [39].
(16)$$\begin{array}{cc} F_{mn,real}^{mid} & =A_{mn}\sin\left(a_{n}\right)\times \mu \left\{fe_{-IA}\right\}\times \mu \left\{\cos\left(\Phi_{mn}\right)\right\}\\ F_{mn,real}^{rad} & =A_{mn}\sin\left(a_{n}\right)\times \left(\begin{array}{c} \left|\mu \left\{fe_{-IA}\right\}\right|\times \frac{1}{2}\omega \left\{\cos\left(\Phi_{mn}\right)\right\}+\left|\mu \left\{\cos\left(\Phi_{mn}\right)\right\}\right|\\ \times \frac{1}{2}\omega \left\{fe_{-IA}\right\}+\frac{1}{2}\omega \left\{fe_{-IA}\right\}\times \frac{1}{2}\omega \left\{\cos\left(\Phi_{mn}\right)\right\} \end{array}\right) \end{array}$$
(17)$$\begin{array}{cc} F_{mn,imag}^{mid} & =A_{mn}\sin\left(a_{n}\right)\times \mu \left\{fe_{-IA}\right\}\times \mu \left\{\sin\left(\Phi_{mn}\right)\right\}\\ F_{mn,imag}^{rad} & =A_{mn}\sin\left(a_{n}\right)\times \left(\begin{array}{c} \left|\mu \left\{fe_{-IA}\right\}\right|\times \frac{1}{2}\omega \left\{\sin\left(\Phi_{mn}\right)\right\}+\left|\mu \left\{\sin\left(\Phi_{mn}\right)\right\}\right|\\ \times \frac{1}{2}\omega \left\{fe_{-IA}\right\}+\frac{1}{2}\omega \left\{fe_{-IA}\right\}\times \frac{1}{2}\omega \left\{\sin\left(\Phi_{mn}\right)\right\} \end{array}\right) \end{array}$$
where μ represents the midpoint, ω represents the interval width, and the subscripts real and imag represent the real and imaginary parts of the complex item, respectively. The interval calculation of $\cos(\Phi_{mn})$ and $\sin(\Phi_{mn})$ are used in the method in [32]; then the upper and lower boundaries of the power pattern interval of CAAs are as indicated in (18) and (19), respectively.
(18)$$P_{Rs}^{\inf}=\left\{ \begin{array}{c} \min\left({\left(F_{real}^{\inf}\right)}^{2},{\left(F_{real}^{\sup}\right)}^{2}\right)+\min\left({\left(F_{imag}^{\inf}\right)}^{2},{\left(F_{imag}^{\sup}\right)}^{2}\right);\\ \;\;\left(F_{real}^{\inf}>0||F_{real}^{\sup}<0\right)\&\&\left(F_{imag}^{\inf}>0||\left(F_{imag}^{\sup}\right)<0\right)\\ \min\left({\left(F_{real}^{\inf}\right)}^{2},{\left(F_{real}^{\sup}\right)}^{2}\right);\\ \;\;\left(F_{real}^{\inf}>0||F_{real}^{\sup}<0\right)\&\&\left(F_{imag}^{\inf}\leq 0||\left(F_{imag}^{\sup}\right)\geq 0\right)\\ \min\left({\left(F_{imag}^{\inf}\right)}^{2},{\left(F_{imag}^{\sup}\right)}^{2}\right);\\ \;\;\left(F_{real}^{\inf}\leq 0||F_{real}^{\sup}\geq 0\right)\&\&\left(F_{imag}^{\inf}>0||\left(F_{imag}^{\sup}\right)<0\right)\\ 0;\;Otherwise \end{array}\right.$$
(19)$$P_{Rs}^{\sup}=\max\left({\left(F_{real}^{\inf}\right)}^{2},{\left(F_{real}^{\sup}\right)}^{2}\right)+\max\left({\left(F_{imag}^{\inf}\right)}^{2},{\left(F_{imag}^{\sup}\right)}^{2}\right)$$

In addition to the structural uncertainty analysis of the element factors in the CAAs, the interval analysis of the excitation error on the electrical performance can also be discussed via the proposed method. Based on (10), the excitation amplitude is taken as the interval variable, as shown in (20). The influence of the tolerance of the excitation amplitude on the electrical performance of the antenna can be further studied.
(20)$$\begin{array}{c} F\left(\theta ,\varphi \right)=\sum_{m=1}^{N}\sum_{n=1}^{N}A_{mn-IA}\times fe_{mn}\left(\theta ,\varphi \right)\times \sin(a_{n})e^{jk_{0}\left(ux_{mn}+vy_{mn}+\cos\theta z_{mn}\right)}\\ A_{mn-IA}=\left[A_{mn}-\delta \times A_{mn},A_{mn}+\delta \times A_{mn}\right] \end{array}$$

Based on (5), the excitation phase is taken as the interval variable, as shown in (21), where $B_{mn}=\left[B_{mn}^{\inf};B_{mn}^{\sup}\right]$ is the phase error of the CAAs, with the upper and low boundaries $B_{mn}^{\inf}=b_{mn}-\beta$ and $B_{mn}^{\sup}=b_{mn}+\beta$, respectively, and *β* is the phase error (unit = degree). Similarly, the interval analysis method RIA in [13] substitutes (21) into the interval formula of the power pattern to obtain the upper and lower boundaries.
(21)$$F\left(\theta ,\varphi \right)=\sum_{m=1}^{N}\sum_{n=1}^{N}A_{mn}fe_{mn}\left(\theta ,\varphi \right)\times \sin(a_{n})e^{j\left[k_{0}\left(ux_{mn}+vy_{mn}+\cos\theta z_{mn}\right)+B_{mn}\right]}$$

Because the phase and radii errors are located in the exponent part, they can also be expressed by Φ_*mn*_, as shown in (22).
(22)$$\Phi_{mn}=k_{0}\left(ux_{mn}+vy_{mn}+\cos\theta z_{mn}\right)+B_{mn}$$

## 3. Numerical Examples of Verification

### 3.1. Verification of Spherical Element Factors

This section tries to verify the surrogate model of the spherical element factor (spherical patch antenna pattern). The patch antenna (as element) is designed and simulated by full-wave method (HFSS) as shown in Figure 7. The radius of the spherical substrate is an interval variable; it is 500 mm with an error of 1.5 mm, the length of the patch is L = 41 mm, the width is W = 41 mm, the distance from the bottom feed to the center is 6.5 mm, the permittivity is 2.2, and the thickness of the substrate is 2 mm. It is assumed that the radius of curvature at any point of the antenna element is the same, there is no radius distribution in the patch, and the radius of the different element antennas can be different. When there is an error in the radius of the spherical substrate, the corresponding element factor is affected but the upper and lower boundaries of all antenna element patterns are consistent.

There are two parameters used for BP neural network training in this research: spherical substrate radius and radiation angle. The spherical substrate radius error samples simulated in HFSS are 31 (498.5 mm–501.5 mm, with an interval of 0.1 mm) and 1801 radiation angle samples (−90°–90°, with an interval of 0.1°). These two samples are combined and 55,831 databases were generated. The database is randomly forked and the ratios of 80%, 10%, and 10% are used for training, validation, and testing, respectively. The comparison between the actual value and the predicted value is shown in Figure 8. When the number of nodes in the middle-hidden layer is 10, the mean square error at this time is 0.0276. Therefore, the prediction of new data within the error range can be made using the trained network.

Taking the E-plane element pattern as an example, the element factor pattern interval obtained by the surrogate model is shown in Figure 9. The green line is the MC simulation result of the trained BP neural network (200 times in total) with radius varies in interval [498.5, 501.5] mm randomly, and the red and blue lines are the upper and lower boundaries fitted by MATLAB. The fitting accuracy of MATLAB was 10^−25^. The normalized element factor interval is listed in Table 1, where Δ represents the interval width. The interval obtained by MATLAB completely envelops all simulation results of the BP.

### 3.2. Verification of Planar Arrays

The power pattern of a planar array obtained by the proposed method was compared with the results of a full-wave method (HFSS). A 10 × 10 array antenna was taken as an example and the radius of the spherical substrate was set to infinity. The calculation results were compared with the array antenna simulated using HFSS software, as shown in Figure 10 and Figure 11.

It can be seen from Figure 10 and Figure 11 that, when the radius of the spherical substrate is sufficiently large, the tow power patterns are almost the same and, the closer they are to the main lobe, the closer they are numerically. At a distance from the main lobe, the two patterns are slightly different. It is undeniable that the surrogate model of the element factor obtained by fitting will also incur an error. In addition, the pattern obtained through the proposed method is asymmetrical in the far-side lobes, mainly because the pattern of the element factor obtained by BP neural network is not completely symmetrical, as shown in Figure 9. Then, the examples in Section A and B can prove that the surrogate model can obtain satisfactory calculation accuracy and can be used for the tolerance analysis with high efficiency rather than a full-wave method such as HFSS.

### 3.3. Verification of Spherical Array

This section verifies the proposed method in a spherical conformal array antenna. A 6 × 6 spherical CAA as shown in Figure 1 was designed and simulated by the full-wave method (HFSS). The radius of the spherical substrate was 1000 mm and three cases were compared with the power pattern calculated by HFSS, as shown in Figure 12. Case 1 (blue line): with reference to the method in [28], the element factor was not included and only the array factor was taken into account. Case 2 (red line): with reference to the method in [34], the element and array factors were considered simultaneously and the element factor used the theoretical formula of a planar patch antenna. Case 3 (black line): using the method proposed in this work, the element and array factors were considered simultaneously and the surrogate model of the element factor was obtained using the HFSS data.

In Figure 12, the pink line represents the results obtained through the full-wave method (HFSS), it can be seen that the power pattern trends of the HFSS results are completely consistent, among which the difference between the simulation results of the HFSS with only an array factor is the largest, particularly in the far lobe region. The element factor has a significant effect on the power pattern. The different between the results of the proposed method and the HFSS results is minimal, which demonstrates the reliability and effectiveness of the proposed method.

### 3.4. Verification of Linear/Planar Arrays Interval Analysis

When the spherical radius is sufficiently large, the curved surface tends toward the plane. When the element number m or n is set to 1, this method is also applicable to linear arrays. To verify the proposed interval analysis method of CAAs, a comparison between a spherical substrate with a sufficiently large radius and the planar/linear array antenna is also one of the verification examples. The tolerance analysis of a linear array was implemented in this section and results obtained by this method are compared with data in [28,30]. The results in the presence of the excitation amplitude and phase tolerance are shown in Figure 13 and Figure 14. A data comparison with the literature [28,30] is shown in Table 2 and Table 3.

In the first example, the Dolph–Chebyshev excitation is adopted and the error of excitation amplitude is 1%, 5%, and 10%, respectively. Comparing the interval width of the proposed method and RIA [29], the two values are very close, which can prove that the tolerance analysis results of the amplitude error interval by the proposed method are reasonable.

In the second example, the sidelobe level (SLL) is −20 dB, the excitation mode is the same as the example in [30], and the phase error is 2 and 5 deg. Comparing the interval width of the proposed method and RIA [30], the two values are almost the same, which proves that the tolerance analysis results of the phase error interval by the proposed method are also reasonable.

### 3.5. Verification of Conformal Arrays Interval Analysis

Because there is no comparable interval analysis example for a spherical CAA, the interval results of the linear array on the spherical surface were compared with the results of the plane linear array and the MC method. The radius of the spherical substrate is 500 mm and its ideal power pattern and the pattern with excitation amplitude and phase error are shown in Figure 15 and Figure 16. Table 2 was compared with the example with the amplitude error interval in [28]. The comparison conditions in the two works are the same, i.e., SLL = −10 dB, Dolph–Chebyshev excitation is applied, and the amplitude error ratios are 1%, 5%, and 10%, respectively. A spherical CAA significantly increased the sidelobe. Comparing the interval width of this method and RIA [28], the two results are of the same order of magnitude, which proves that the data calculated by the proposed method are reasonable.

Table 3 was compared with the example with the phase error interval in [30]. The comparison conditions in the two works are the same, i.e., SLL = −20 dB, Dolph–Chebyshev excitation is applied, and the phase error is 2 deg and 5 deg. The interval width between the method in this study and RIA [31] was compared. Although the interval width obtained by the proposed method is less than that of the RIA [31], it is still of the same order of magnitude. Therefore, it can be proved that the phase error data calculated by the proposed method are reasonable.

## 4. Tolerance Analysis of Spherical CAAs with Error Interval

In this section, a tolerance analysis carried out for the spherical CAAs with radius, excitation amplitude, and phase error intervals is described.

### 4.1. Tolerance Analysis of Spherical CAA with Radius Error Interval

The 6 × 6 array antenna in Figure 1 was also taken as an example. Under uniform excitation, the spherical substrate radius is 500 mm and the error of the radius error is {±0.5, ±1.0, ±1.5} mm. The selected 6 × 6 array is completely symmetrical and the patterns of the E- and H-planes are roughly the same. Therefore, the analysis results of the E-plane power pattern are shown in Figure 17 and Figure 18 and compared with the results of 20,000 Monte Carlo simulations. The results are listed in Table 4. In addition, we also used the Ruze formula ($G=G_{0}e^{-{\left(4\pi \delta_{rms}/\lambda \right)}^{2}}$) to calculate the gain deviation under different error conditions for comparison with the interval gain pattern calculated by this study method [16]. When the errors are 0.5 mm, 1 mm, and 1.5 mm, the gain deviations calculated by the Ruze formula are −0.042 dB, −0.168 dB, and −0.379 dB, respectively, while the deviation intervals obtained in this study are [−0.66; 0.66] dB, [−1.05; 1.02] dB, and [−1.52; 1.37] dB. The peak deviation estimated by the Ruze formula are all within the interval obtained by this study method.

### 4.2. Tolerance Analysis of Spherical CAA with Excitation Amplitude Error Interval

Using the same spherical CAAs with a uniform excitation and an amplitude error of δ = {1%, 3%, 5%} and comparing via Monte Carlo results (20,000 times), the interval results of the E-plane power pattern of the spherical CAA were obtained, as shown in Figure 19 and Figure 20 and listed in Table 5.

### 4.3. Tolerance Analysis of Spherical CAA with Excitation Phase Error Interval

A uniform excitation occurred, $b_{mn}=0$, and an error of β{1,2,3}deg was achieved. Compared with the results of the Monte Carlo method, the interval results of the CAA are shown in Figure 21 and Figure 22 and the results are listed in Table 6.

For the spherical CAAs, the following conclusions can be drawn from the above three numerical examples: (1) when the radius is an interval variable, the width of the interval is the largest; (2) when the MC simulation result is compared with the power pattern, the element factor adopts the surrogate model and the RIA method adopted by the array factor is feasible and the simulation result is valid; (3) all examples verify the envelope of the interval. The premise of the interval analysis method used in this section was to have an explicit expression of the power pattern and to allow some of the variables to be intervalized in order to obtain an intervalized result.

### 4.4. Spherical CAAs with Different Radii

Taking a linear array of 1 × 10 as an example, the influence of the curvature radius of the substrate on the electrical performance was calculated through MATLAB when the spherical conformation antenna was studied, as shown in Figure 23. The black dotted line in the figure is the pattern of plane E of the planar array and the other curves are the pattern of the spherical antenna with a radius from 400 mm to 3000 mm.

Through a comparison between the patterns of spherical array antennas with different radii and planar array antennas with an increase in radius, the beam width of the power pattern decreases and the pattern of the spherical antenna gradually approaches the pattern of the planar array antenna, which verifies the correctness of the derived expression of the spherical conformal array antenna. In addition, the derived expression can also be used for any array within a spherical range. Examples are shown in Figure 24.

### 4.5. Influence of Element and Array Factors on Interval of Array Antenna Pattern

In this section, with spherical substrates of the same size (Rs = 500 mm) and error (±0.5 mm), the power patterns of a 1 × 10 linear array and 10 × 10 planar array antennas were analyzed using a MATLAB simulation under different conditions to explain the interval influence of the element and array factors on the pattern of the array antenna. As shown in Figure 25 and Figure 26, all curves were calculated using MATLAB. The black line represents the ideal array antenna pattern, the green and red lines are the interval results obtained when only element or array factor errors occur, respectively, and the blue line represents the pattern interval when both the array factor and the array factor error occur simultaneously. It can be seen from the figure that, for both a linear array and a planar array, the pattern interval in which the element and array factor errors occur at the same time envelops the result in which only the array factor error occurs, and the pattern interval in which only the array factor error occurs envelops the result in which only the element factor error occurs.

According to the analysis of the results in Table 7 and Table 8, when an error occurs in the radius of the spherical dielectric substrate, the array factor error (coordinate position error of the antenna unit) has a greater impact on the array antenna pattern and the element factor error has a smaller impact on the array antenna pattern but cannot be ignored.

## 5. Conclusions

This study follows the surrogate model method for an element factor and the interval calculation method for the real and imaginary parts of an array factor. The following conclusions can be drawn from the analysis of the power pattern interval of the non-expandable spherical CAA:

(1) When the radius is an interval variable, the interval width is the largest; thus, the proportion of the interval radius is reduced, the remaining error is a few percentage points, and the radius is a few tenths of a percent. 

(2) Compared with the Monte Carlo method, the method used in an interval analysis can effectively, simply, and reasonably solve the influence of the spherical substrate radius error and the excitation amplitude and phase error on the pattern.

(3) The object of this article is a spherical CAA that has not been analyzed for the interval and provides new material for the development of an interval analysis in the field of antennas.

(4) For explicit expressions that are difficult to derive directly, surrogate models or machine learning algorithms can be used to solve them. This research uses the BP intelligent algorithm to build the surrogate model; other artificial intelligence algorithms are also suitable for this research.

## Figures and Tables

**Figure 1 sensors-22-09828-f001:**
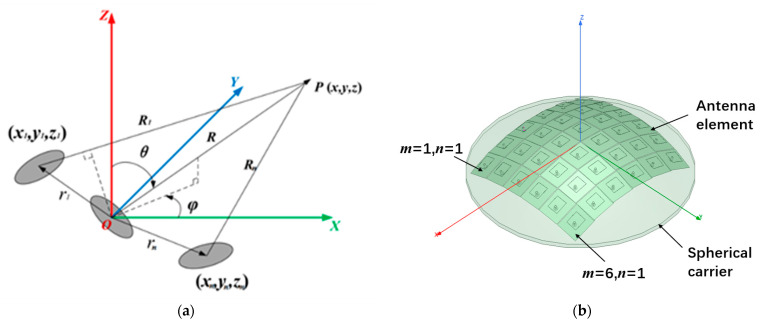
Simple model and spatial placement model of 3D conformal array antenna ((**a**) diagram of space conformal array antenna, (**b**) model diagram established by HFSS).

**Figure 2 sensors-22-09828-f002:**
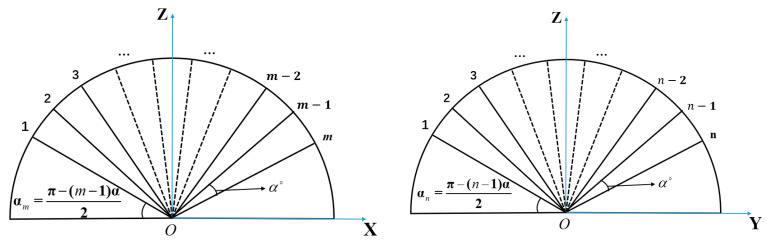
*m × n* spherical CAA angle calculation model.

**Figure 3 sensors-22-09828-f003:**
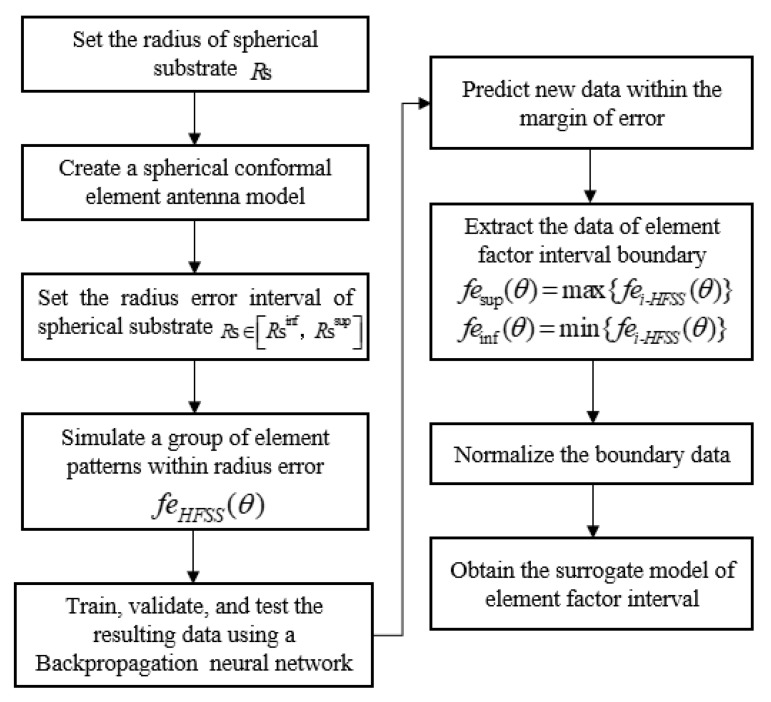
The flow chart of the surrogate model.

**Figure 4 sensors-22-09828-f004:**
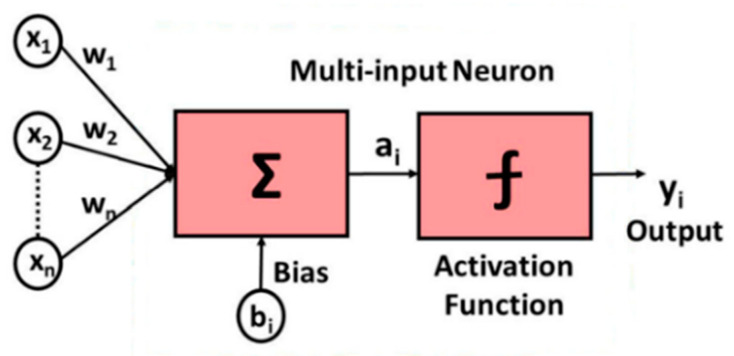
Basic structure of a neuron.

**Figure 5 sensors-22-09828-f005:**
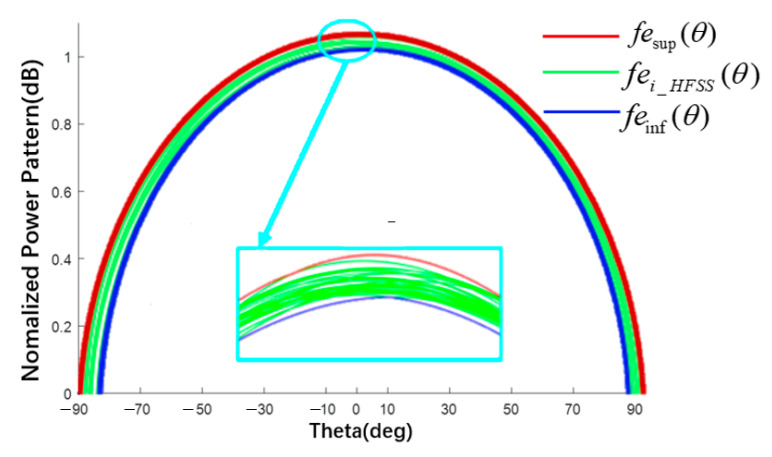
Element pattern (schematic of surrogate model for obtaining the element factor interval).

**Figure 6 sensors-22-09828-f006:**
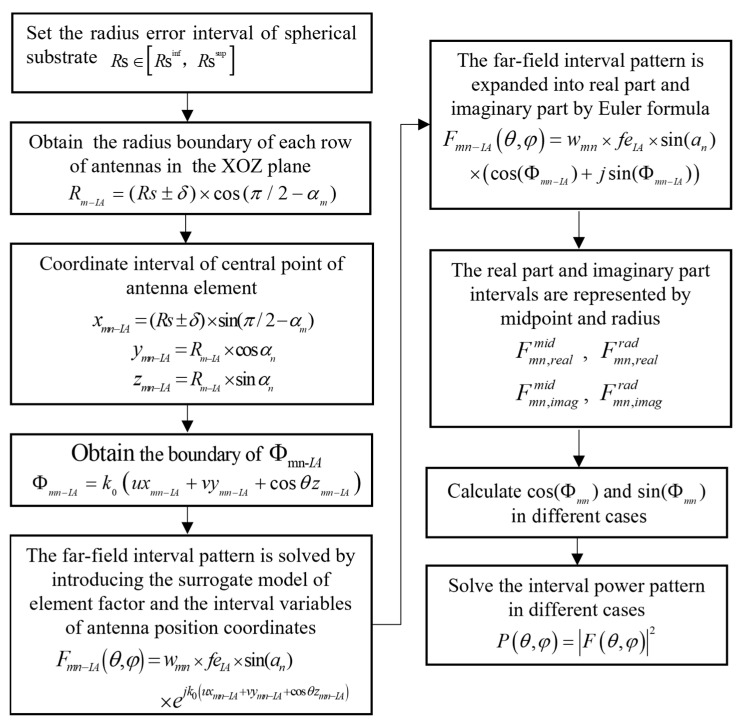
The flow chart for solving the interval power pattern.

**Figure 7 sensors-22-09828-f007:**
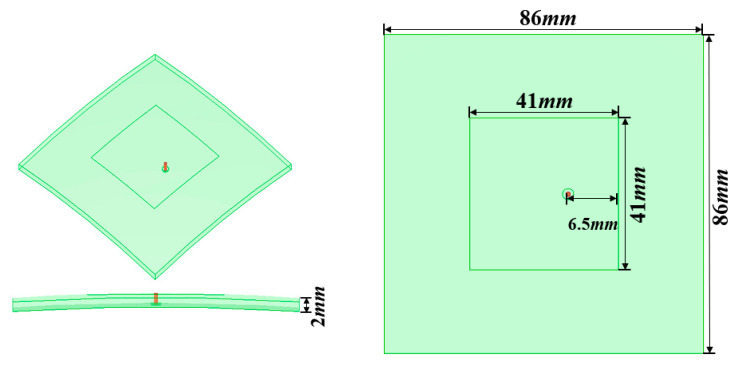
Model of conformal patch antenna.

**Figure 8 sensors-22-09828-f008:**
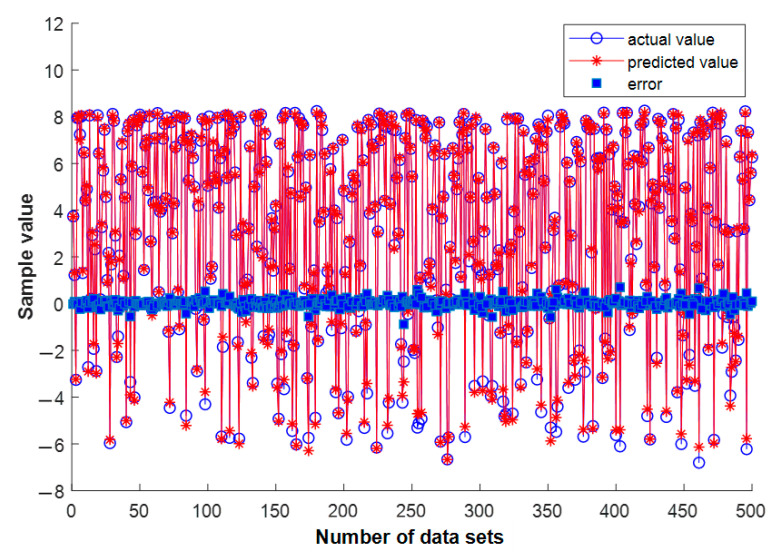
Comparison of predicted value and actual value of BP neural network test set.

**Figure 9 sensors-22-09828-f009:**
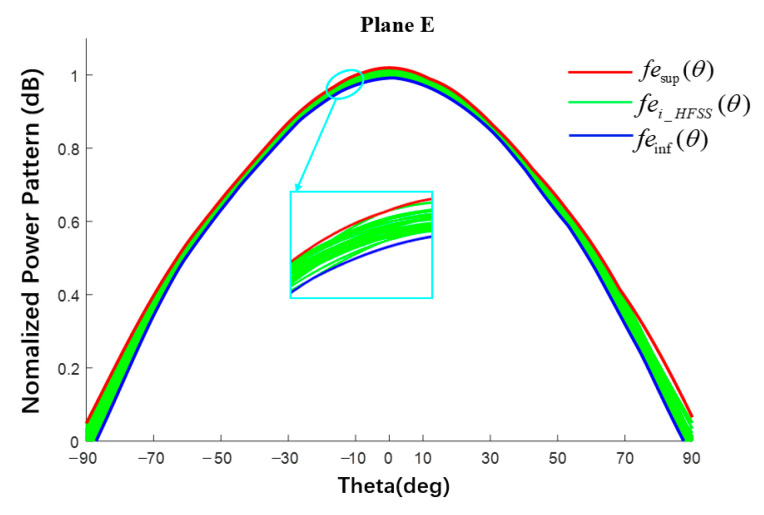
Interval pattern of E-plane element factor.

**Figure 10 sensors-22-09828-f010:**
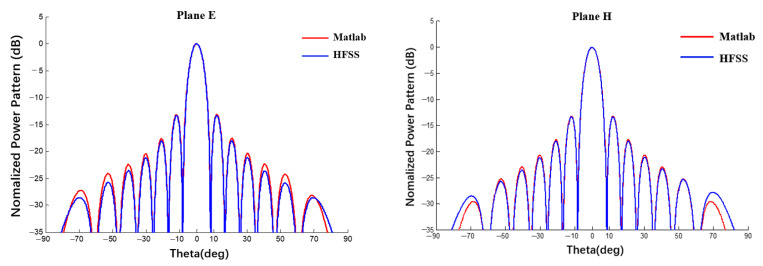
Simulation of 10 × 10 array antenna contrast pattern using MATLAB and HFSS (uniform excitation).

**Figure 11 sensors-22-09828-f011:**
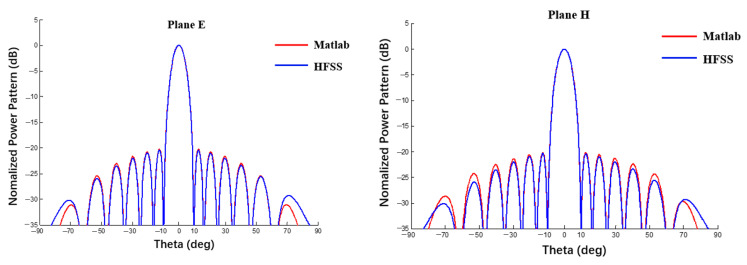
Simulation of 10 × 10 array antenna contrast pattern using MATLAB and HFSS (Dolph–Chebyshev excitation).

**Figure 12 sensors-22-09828-f012:**
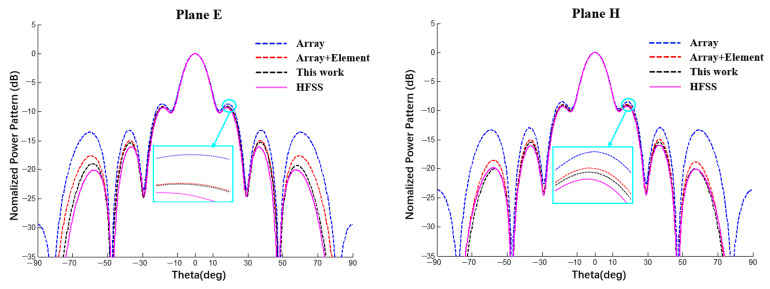
Comparison of simulation results between HFSS and MATLAB (Plane E, uniform excitation).

**Figure 13 sensors-22-09828-f013:**
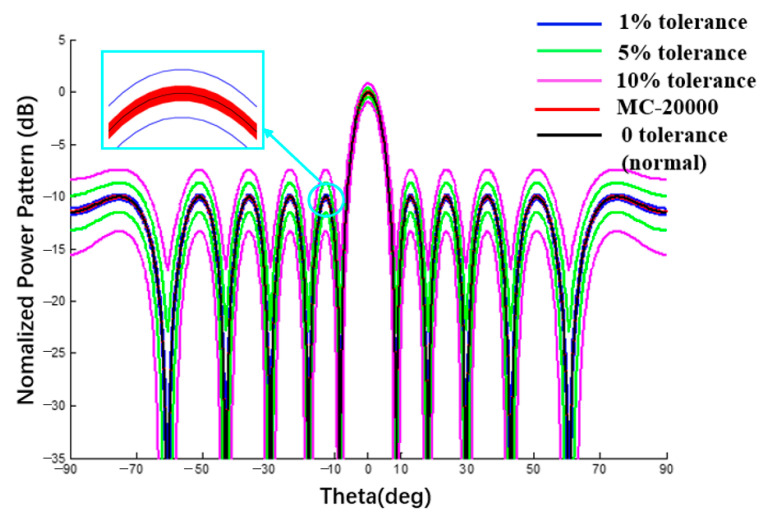
Linear array antenna pattern with excitation amplitude tolerance (Plane E, Dolph–Chebyshev excitation, the tolerance of MC method is 1%).

**Figure 14 sensors-22-09828-f014:**
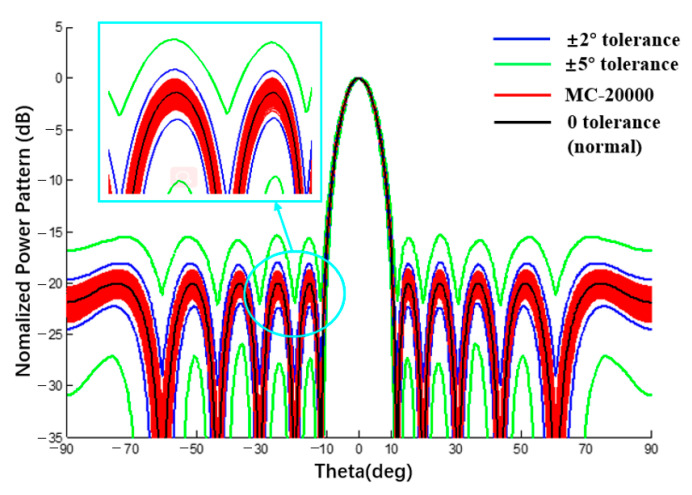
Linear array antenna pattern with excitation phase tolerance (Plane E, Dolph–Chebyshev excitation, the tolerance of MC method is ±2°).

**Figure 15 sensors-22-09828-f015:**
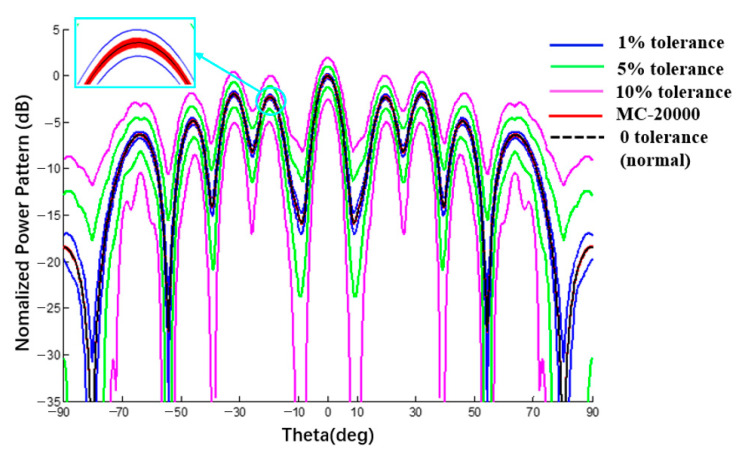
Power pattern under different excitation amplitude tolerances (Plane E, Dolph–Chebyshev excitation, the tolerance of MC method is 1%).

**Figure 16 sensors-22-09828-f016:**
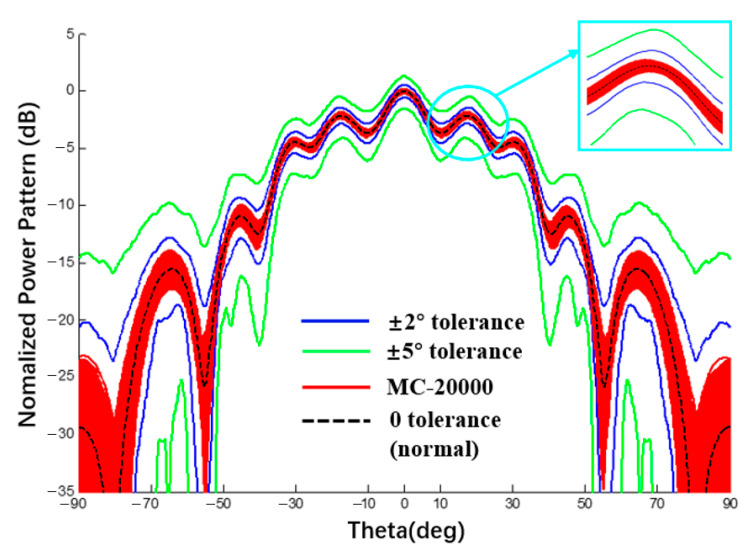
Power pattern under different excitation phase tolerances (Plane E, Dolph–Chebyshev excitation, the tolerance of MC method is ±2°).

**Figure 17 sensors-22-09828-f017:**
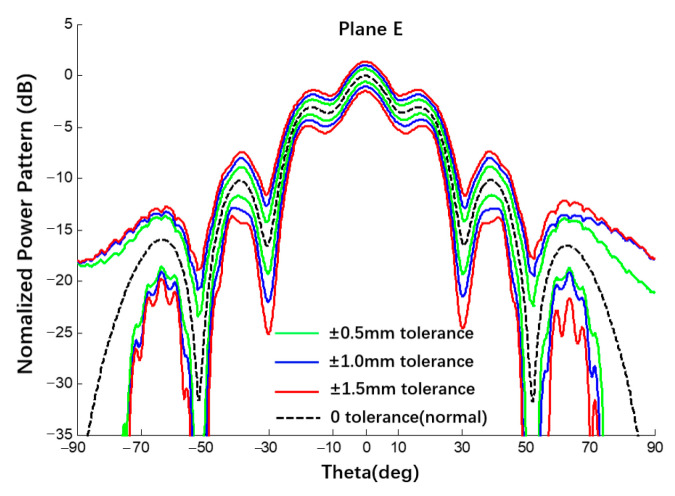
Interval power pattern of radius error of spherical CAAs (Plane E, uniform excitation).

**Figure 18 sensors-22-09828-f018:**
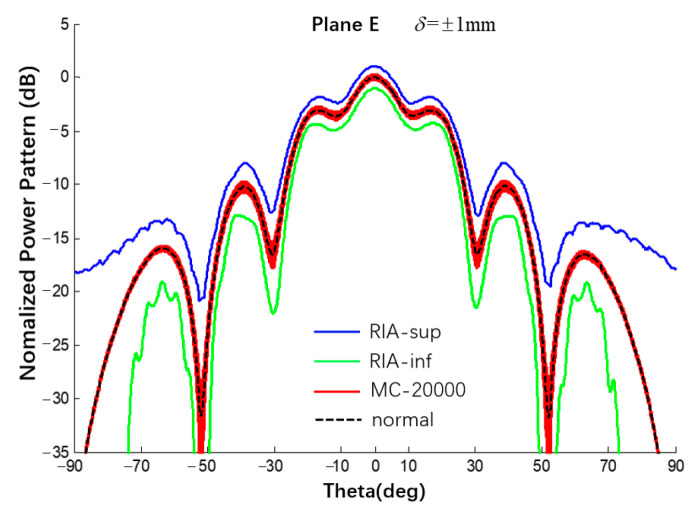
Interval power pattern of radius error of spherical CAAs (Plane E, uniform excitation, the tolerance of MC method is ±1 mm).

**Figure 19 sensors-22-09828-f019:**
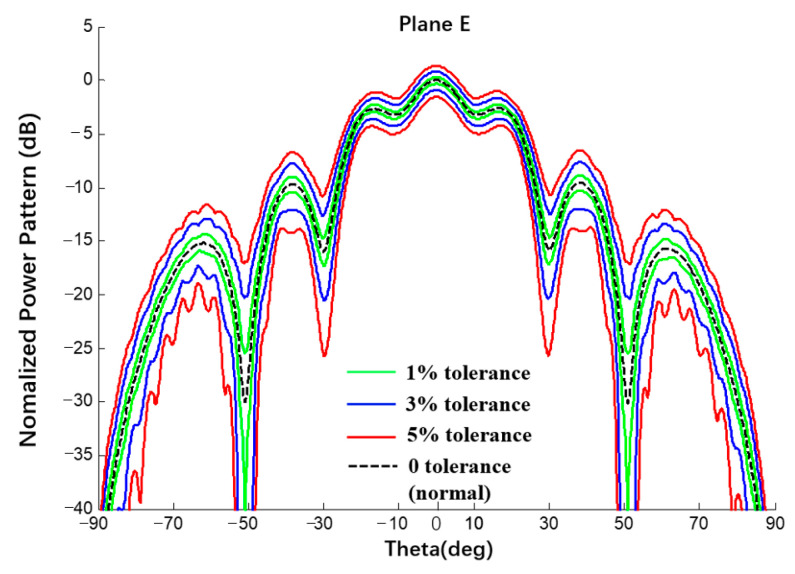
Interval power pattern of excitation amplitude error of spherical CAAs (Plane E, uniform excitation).

**Figure 20 sensors-22-09828-f020:**
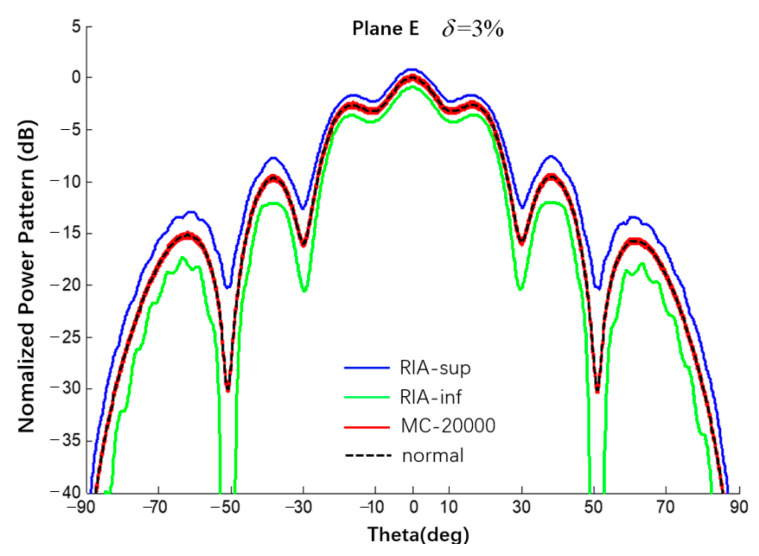
Interval power pattern of excitation amplitude error of spherical CAAs (Plane E, uniform excitation, the tolerance of MC method is 3%).

**Figure 21 sensors-22-09828-f021:**
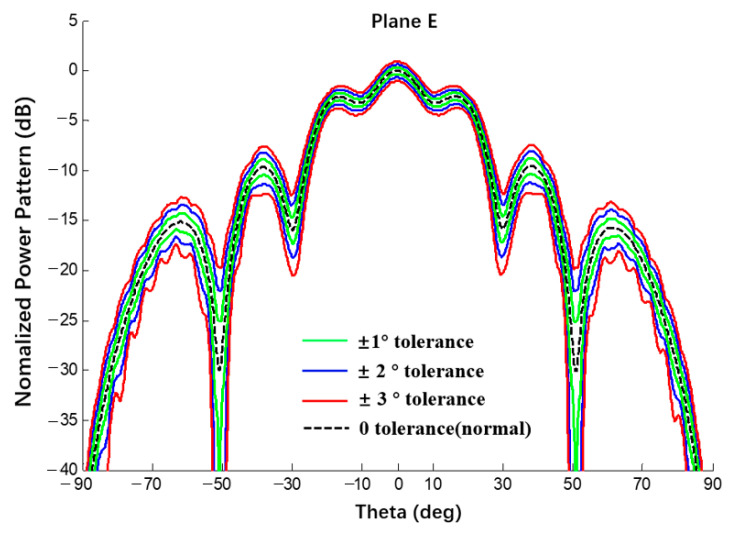
Interval power pattern of excitation phase error of spherical CAAs (Plane E, uniform excitation).

**Figure 22 sensors-22-09828-f022:**
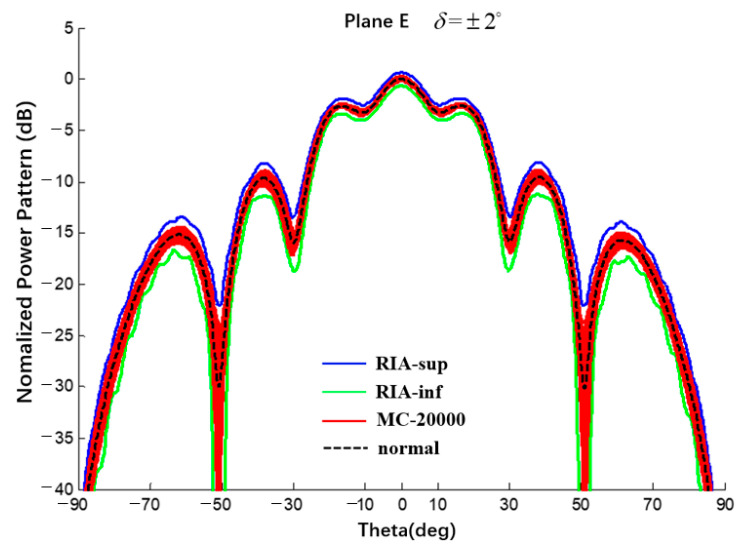
Interval power pattern of excitation phase error of spherical CAAs (Plane E, uniform excitation, the tolerance of MC method is ±2°).

**Figure 23 sensors-22-09828-f023:**
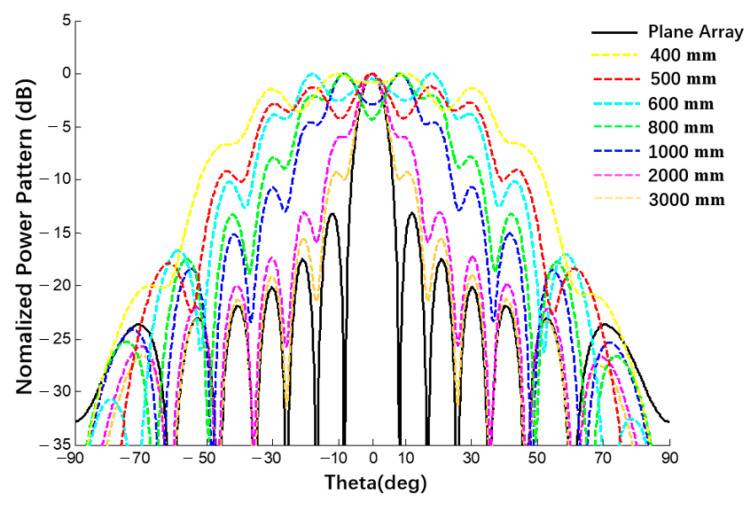
Comparison of the influence of different radii on the pattern when calculated using MATLAB (uniform excitation).

**Figure 24 sensors-22-09828-f024:**
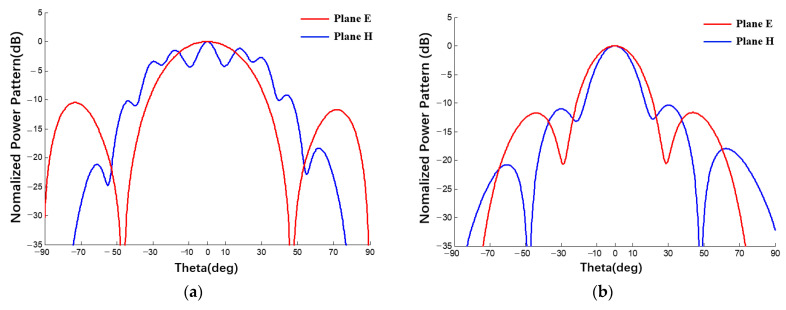
Power patterns of spherical conformal array antennas with different specifications: (uniform excitation, (**a**) 2 × 10, (**b**) 3 × 4, (**c**) 4 × 6, (**d**) 5 × 8).

**Figure 25 sensors-22-09828-f025:**
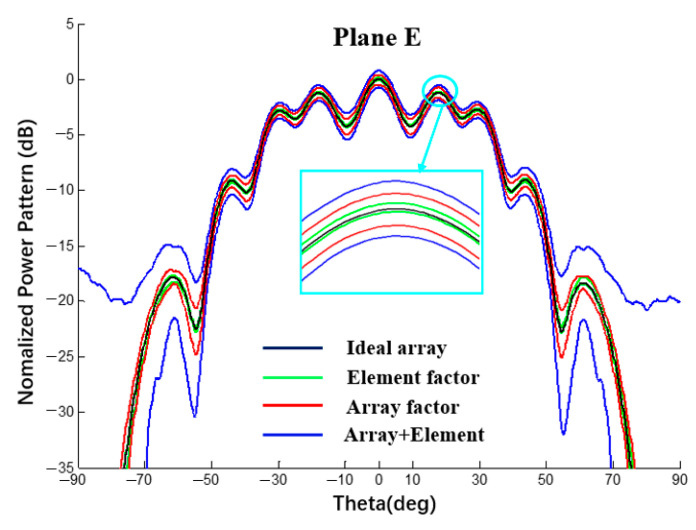
Interval power pattern based on the error of different factors of 1 × 10 linear arrays (uniform excitation).

**Figure 26 sensors-22-09828-f026:**
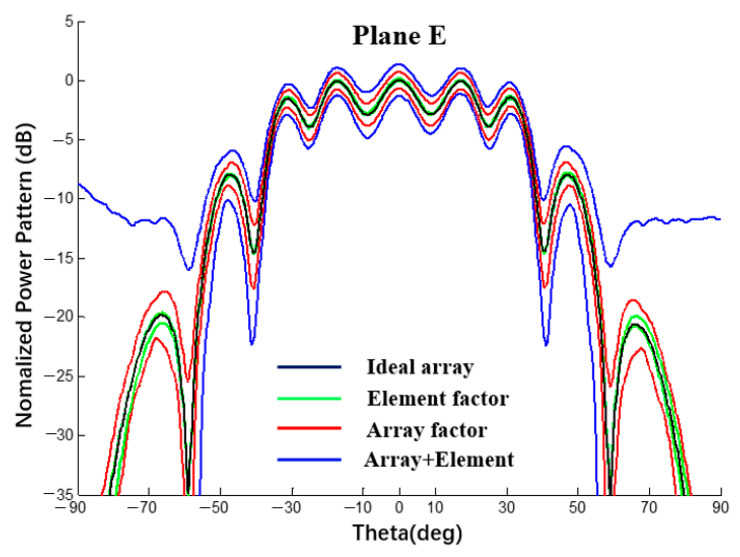
Interval power pattern based on the error of different factors of 10 × 10 arrays (uniform excitation).

**Table 1 sensors-22-09828-t001:** Interval value of element factor of plane E in Figure 9.

Theta (Deg)	BP Results	Δ	This Work	Δ
−60	[0.507; 0.540]	0.033	[0.507; 0.541]	0.034
−30	[0.847; 0.868]	0.021	[0.843; 0.870]	0.027
0	[0.993; 1.015]	0.022	[0.991; 1.020]	0.029
30	[0.850; 0.869]	0.019	[0.850; 0.870]	0.020
60	[0.489; 0.536]	0.047	[0.487; 0.541]	0.054

**Table 2 sensors-22-09828-t002:** Numerical results of power pattern in Figure 13 and Figure 15.

SLL(dB)	Amplitude Error	Line Array (Figure 13)	ΔL	Curvilinear Array (Figure 15)	ΔC	[28]	Δ
−10	1%	[−10.28; −9.64]	0.64	[−3.08; −2.58]	0.50	[−10.29; −9.68]	0.61
	5%	[−11.51; −8.42]	3.09	[−4.15; −1.62]	2.53	[−11.49; −8.43]	3.06
	10%	[−13.11; −6.89]	6.22	[−5.56; −0.53]	5.03	[−13.10; −6.96]	6.14

**Table 3 sensors-22-09828-t003:** Numerical results of power pattern in Figure 14 and Figure 16.

SLL (dB)	Phase Error	Line Array (Figure 14)	ΔL	Curvilinear Array (Figure 16)	ΔC	[30]	Δ
−20	2°	[−21.06; −18.92]	2.14	[−3.30; −1.91]	1.39	[−20.58; −18.42]	2.16
	5°	[−26.30; −15.97]	10.33	[−4.50; −0.99]	3.51	[−25.45; −14.81]	10.64

**Table 4 sensors-22-09828-t004:** Numerical results of SLL of power pattern in Figure 17 and Figure 18.

Radius Error		±0.5 mm	Δ0.5	±1.0 mm	Δ1.0	±1.5 mm	Δ1.5
Main 0 dB	This Work	[−0.66; 0.66]	1.32	[−1.05; 1.02]	2.07	[−1.52; 1.37]	2.89
	MC	[−0.13; 0.14]	0.27	[−0.23; 0.28]	0.51	[−0.38; 0.42]	0.80
1st −3.09 dB	This Work	[−3.80; −2.35]	1.45	[−4.34; −1.87]	2.47	[−4.90; −1.42]	3.48
	MC	[−3.26; −2.92]	0.34	[−3.40; −2.74]	0.66	[−3.57; −2.60]	0.97

**Table 5 sensors-22-09828-t005:** Numerical results of SLL of power pattern in Figure 19 and Figure 20.

Amplitude Error		1%	Δ1%	3%	Δ3%	5%	Δ5%
Main 0 dB	This Work	[−0.29; 0.28]	0.57	[−0.90; 0.83]	1.73	[−1.55; 1.34]	2.89
	MC	[−0.09; 0.09]	0.18	[−0.26; 0.26]	0.52	[−0.44; 0.43]	0.87
1st −2.63 dB	This Work	[−2.95; −2.31]	0.64	[−3.62; −1.68]	1.94	[−4.29; −1.08]	3.21
	MC	[−2.71; −2.54]	0.17	[−2.89; −2.37]	0.52	[−3.07; −2.20]	0.87

**Table 6 sensors-22-09828-t006:** Numerical results of SLL of power pattern in Figure 21 and Figure 22.

Phase Error		1 Deg	Δ1 Deg	2 Deg	Δ2 Deg	3 Deg	Δ3 Deg
Main 0dB	This Work	[−0.33; 0.31]	0.64	[−0.67; 0.61]	1.28	[−1.03; 0.90]	1.93
	MC	[−0.13; 0.13]	0.26	[−0.26; 0.26]	0.52	[−0.41; 0.38]	0.79
1st −2.63dB	This Work	[−3.01; −2.25]	0.76	[−3.41; −1.89]	1.52	[−3.81; −1.54]	2.27
	MC	[−2.75; −2.52]	0.23	[−2.84; −2.42]	0.42	[−2.94; −2.32]	0.62

**Table 7 sensors-22-09828-t007:** SLL interval value in Figure 25 (1 × 10 linear array).

Lobe	Array + Element Tolerance	ΔAE	Array Factor Tolerance	ΔA	Element Factor Tolerance	ΔE	ΔA/ΔAE	ΔE/ΔAE
Main	[−0.76; 0.77]	1.53	[−0.46; 0.43]	0.89	[−0.08; 0.16]	0.24	58.2%	15.7%
1st	[−1.94; −0.52]	1.42	[−1.71; −0.79]	0.92	[−1.29; −1.11]	0.18	64.8%	12.7%
2nd	[−3.51; −2.11]	1.40	[−3.26; −2.39]	0.87	[−2.84; −2.69]	0.15	62.1%	10.7%
3rd	[−10.41; −8.11]	2.30	[−9.75; −8.66]	1.09	[−9.31; −9.08]	0.23	47.4%	10.0%
4th	[−21.55; −14.94]	6.61	[−18.41; −17.20]	1.21	[−18.29; −17.68]	0.61	18.3%	9.2%

**Table 8 sensors-22-09828-t008:** SLL interval value in Figure 26 (10 × 10 array).

Lobe	Array + Element Tolerance	ΔAE	Array Factor Tolerance	ΔA	Element Factor Tolerance	ΔE	ΔA/ΔAE	ΔE/ΔAE
Main	[−1.33; 1.32]	2.65	[−0.68; 0.68]	1.36	[−0.08; 0.16]	0.24	51.3%	9.1%
1st	[−1.26; 1.08]	2.34	[−0.80; 0.59]	1.39	[−0.16; 0.03]	0.19	59.4%	8.1%
2nd	[−2.90; −0.27]	2.63	[−2.30; −0.82]	1.48	[−1.61; −1.42]	0.19	56.3%	7.2%
3rd	[−10.12; −5.92]	4.20	[−8.90; −6.98]	1.92	[−8.10; −7.86]	0.24	45.7%	5.7%

## Data Availability

Not applicable.

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
