# Peer review of "Surrogate-Model-Based Interval Analysis of Spherical Conformal Array Antenna with Power Pattern Tolerance"

_sensors, 2022, doi:10.3390/s22249828_

Round 1

Reviewer 1 Report

The analysis in the paper started with Eq. (1). However, Eq. (1) assumes that all the element patterns in an array are the same. In a conformal array, this assumption is invalid, thus rendering the analysis incorrect. 

Author Response

Dear Reviewer, I have replied to your comments, please see the attachment. Thank you and have a good day!

Reviewer 2 Report

The focus of the authors' work is an important area in conformal antenna array design, viz., the effects of tolerance in the antenna structure design including amplitude, phase and curvature radii of the array. 

Suggestions are:

* Perhaps the abstract could be improved, with a line or two to explain the importance and application of this work in conformal area design

* The aim of the work focused on modeling of practical design tolerances and their effects on radiation pattern.  Hence, the basic mathematical equations for the electric/radiation field of a practical array element (such as a patch that the authors mention) should be included.

* The antenna array radiation, described in eqn. (1) is generally correct; however, for antenna sources with vector currents such as patch antennas, the equation would be inherently more complex. This should be elaborated by the authors, since spherical geometry is more complex in analysis than planar geometry, where all element currents can be assumed to have the same direction of polarization. Point sources might be an exception, which are however not practical design elements.

* The authors report several simulation examples to describe the tolerance effects simulation, which is quite appropriate and comprehensive; however, the basic mathematical theory of a spherical antenna structure should be clearly explained, prior to simulation studies.

Author Response

Dear reviewers,

I have responded to your comments point by point, please refer to the attachment. Thank you and have a nice day!

Round 2

Reviewer 1 Report

Thanks for addressing for feedback.

Author Response

Dear reviewer, the revised manuscript has been greatly benefited from the constructive comments and valuable suggestions. We would like to thank you sincerely for the time and effort spent to help improve the presentation of this paper.

Reviewer 2 Report

The authors have reviewed and modified the document, as per the reviewer's queries and suggestions.

Author Response

Dear reviewer, we would like to express our sincerest gratitude to you. Your valuable comments and suggestions are constructive to our work. We deeply appreciate the plenty of time and effort spent to help us improve the paper. Thanks to your preciseness, patience, carefulness, and kindness, the English writing and scientific of this paper have been improved greatly.